# Unveiling the Potential of Droplet Generation, Sorting, Expansion, and Restoration in Microfluidic Biochips

**DOI:** 10.3390/mi10110756

**Published:** 2019-11-06

**Authors:** Yi-Lung Chiu, Ruchi Ashok Kumar Yadav, Hong-Yuan Huang, Yi-Wen Wang, Da-Jeng Yao

**Affiliations:** 1Department of Power Mechanical Engineering, National Tsing Hua University, Hsinchu 30013, Taiwan; johnnychiou2008@gmail.com (Y.-L.C.); ruchiakyadav@gmail.com (R.A.K.Y.); 2Department of Obstetrics and Gynecology, Chang Gung Memorial Hospital, Taoyuan 33305, Taiwan; iwen0711@gmail.com; 3Department of Obstetrics and Gynecology, Chang Gung University and College of Medicine, Taoyuan 33305, Taiwan; 4Institute of NanoEngineering and MicroSystems, National Tsing Hua University, Hsinchu 30013, Taiwan

**Keywords:** biochip, continue microfluidic system, droplet formation, droplet expansion and restoration, and embryo culture

## Abstract

Microfluidic biochip techniques are prominently replacing conventional biochemical analyzers by the integration of all functions necessary for biochemical analysis using microfluidics. The microfluidics of droplets offer exquisite control over the size of microliter samples to satisfy the requirements of embryo culture, which might involve a size ranging from picoliter to nanoliter. Polydimethylsiloxane (PDMS) is the mainstream material for the fabrication of microfluidic devices due to its excellent biocompatibility and simplicity of fabrication. Herein, we developed a microfluidic biomedical chip on a PDMS substrate that integrated four key functions—generation of a droplet of an emulsion, sorting, expansion and restoration, which were employed in a mouse embryo system to assess reproductive medicine. The main channel of the designed chip had width of 1200 μm and height of 500 μm. The designed microfluidic chips possessed six sections—cleaved into three inlets and three outlets—to study the key functions with five-day embryo culture. The control part of the experiment was conducted with polystyrene (PS) beads (100 μm), the same size as the murine embryos, for the purpose of testing. The outcomes of our work illustrate that the rate of success of the static droplet culture group (87.5%) is only slightly less than that of a conventional group (95%). It clearly demonstrates that a droplet-based microfluidic system can produce a droplet in a volume range from picoliter to nanoliter.

## 1. Introduction

Recently, microfluidic chips have attracted much attention from both academia and industry, which has led to their rapid development in the field of medical diagnostics. Microfluidics involve a volume of fluid in the range of microlitre to picolitre, and demonstrate many advantages such as rapid mass delivery and heat transfer, and diminished use of reagents and generation of waste. Miniaturized biological assays or processes on a chip have emerged as a prospective technology. A droplet-based microfluidic system produces a droplet of volume typically ranging from picolitre [1,2] to nanolitre [3,4]. There are two main methods to control the direction or motion of a droplet: one uses a pneumatic valve and the other one employs light waves of a specific wavelength to impinge on a droplet in the channel [5,6,7,8]. The particle-encapsulated droplet can be determined, while droplets absorb the reflected light of a particular spectral wavelength.

Droplet-based microfluidics are commonly conducted with a generating frequency > 10 Hz and a small volume of droplet (picolitre to nanolitre), but have the disadvantages of tedious fabrication, complicated experimental operation, and a requirement of expensive equipment. Embryos are statically cultured from zygote to blastocyst in a 5-μL human tubal fluid (HTF) medium or KSOM medium (Sigma-Aldrich, St. Louis, MO, USA) and covered with culture oil. In reality, the embryos grow dynamically in the oviduct of a female mammal for 5–6 days before implantation. The use of a culture medium as a dispersed phase droplet to encapsulate a mouse embryo flowing in a microchannel for a dynamic culturing condition is thus a highly suitable scheme. 

Some methods to sort droplets use a pneumatic valve as a block to control the direction of motion of the droplet [5,6]. Some contemporary methods to sort droplets apply light waves of a specific wavelength to strike a droplet in the channel. Whether the droplet encapsulates a particle or not is then determined on absorbing reflected light at a particular spectral wavelength. If a droplet encapsulating a particle arrives, the spectrometer transmits a signal to the voltage amplifier that releases a voltage to the electrode. When it is necessary to inject liquid into a droplet, an electric field must be created by the electrode beside the droplet to invalidate temporarily the effect of the surfactant [9,10]. Although dozens of droplets could be sorted and injected in a brief interval, the tedious fabrication, complicated experimental operation, and necessity of many expensive pieces of equipment would be extremely inconvenient.

Many researchers have conducted fundamental studies on droplet microfluidics. Sarvothaman et al. developed a strategy that involved using fluoroalkyl polyethylene glycol copolymers to suppress protein adhesion, which causes a droplet movement to fail [11]. Seiffert et al. proposed more rapid production of droplets by delayed addition of the surfactant to push droplet-based microfluidics to an industrially relevant scale [12]. Pirbodaghi et al. developed an accurate approach that entailed using a bright-field microscope with illumination of white light and a standard high-speed camera to study the fluid dynamics of rapid processes within microfluidic devices [13]. Musterd and coworkers calculated the volume of elongated droplets in microchannels from a top-view image in interpreting experiments on reaction kinetics and transport phenomena [14].

In this work, we developed a microfluidic chip on a polydimethylsiloxane (PDMS) substrate to upgrade the volume of droplets from picolitre and nanolitre to microlitre for culturing mouse embryos. We also devised a convenient technique of operation to attain an effective and stable way for droplet sorting, liquid injection, and liquid exchange.

## 2. Materials and Methods

A microfluidic chip was fabricated using the traditional process. The microfluidic chip was designed to test four main functions of channel performed manually using microfluidic system. There are two main types of medium used for the experiment, the human tubal fluid medium (HTF or KSOM) and the cultural oil medium (OVOIL), which are used as dispersed phase liquid and as continuous phase liquid in the experiment. PDMS is used as main material in fabrication of the channel, due to its favorable mechanical properties like optical transparency, high biocompatibility, high transmittance, and ease of fabrication. Teflon was used after the bonding process to provide a hydrophobic surface coating on the chip [15,16], because the material has a superior biological property and high biocompatibility, which can provide better environmental conditions for embryo development.

Figure 1 shows an overview image of the chip used in the experiment. As indicated in panel (a), the chip has a hexagonal structure with upper base length 64.9 mm, lower base length 28.1 mm, right-hand side length 25.1 mm, and left-hand side length 41.0 mm. As shown in panel (b), the main channel, continuous phase channel, and dispersed phase channel have widths of 1200, 600 and 300 μm, respectively. The channel height is 500 μm.

The chip includes four main segments—a generating area, a sorting area, an enlargement area and a replacement area, along with a storage area and a waste area (see Figure 1c). The chip can use different inlets and outlets, and can also provide different flow rates according to the medium. We can apply multiple medium exchanges of droplets while performing the experiment, but in this experiment, the functions of the chip are set/secured. The mouse embryos are injected via pipette in embryo inlet, using microfluidic system droplets created in the generating area and sorted in the sorting area simultaneously. Later, the droplets without embryos were abandoned to the waste area outlet, whereas the droplets with embryos moved on to the enlargement area. The droplets in the enlargement area are then expanded. The medium from the droplet can eventually be absorbed in the replacement area. 

The chip consists of the following regions—oil inlet, embryo inlet, waste area outlet, enlarge area inlet, replacement area outlet, and storage area outlet. The culture oil flows through the chip via the oil inlet; the culture medium with particles (mouse embryos or PS beads) flows into the chip through the embryo inlet. The abandoned droplets are discarded through the waste area outlet. The HTF or KSOM medium is injected into the droplet to enlarge its size to 5 μL through the enlargement area. The old media are carried away in the replacement area of the chip.

For the fabrication of the wafer, we used a soft lithographic process to fabricate the mold of the channel. Negative photoresist SU-8 (2150) was used on a silicon wafer of thickness less than 300 μm, as shown in the datasheet. We used spin speed of 500 rpm for 40 s. The average thickness of the channel was about 500 μm. During soft baking, the viscosity of the photoresist increased, resulting in a problem of non-uniform thickness of the photoresist. To solve this problem, we rotated the wafer 90° per 3 min during soft baking to ensure uniformity. After spin coating, the ensuing procedure consisted of soft baking, exposure to UV light, and post-exposure baking. The last step involved putting the wafer into the SU-8 developer to wash away the excess photoresist. To prepare the top layer, the tubes were put at the entrance of the mold; then liquid polydimethylsiloxane (PDMS) (A:B = 10:1) was poured on the mold. The liquid PDMS was baked at 85 °C for 30 min until the PDMS became solid. The PDMS was removed from the mold to complete the top layer part of the channel. To enclose the whole channel with PDMS, we coated the glass with PDMS as a bottom layer at spin speed of 2000 rpm for 20 s. The thickness of the PDMS on the glass was about 50 μm [17]. The top and bottom layers were bonded to each other with oxygen plasma to complete fabrication of the chip. We then baked the chip at 70 °C for 10 min to enhance the bonding force. The detailed procedure of fabrication and bonding is shown in Figure 2a.

The breaking methods of the continuous phase liquid to cut off the dispersed phase liquid can be divided into two types—the push-push system and the push-pull system. In the former system, the continuous phase and dispersed phase liquids are both pushed with a forward force, whereas in the latter, the continuous phase liquid is pushed, while the dispersed phase liquid is pulled through the end of the channel (see Figure 2b,c). We herein adopted the push-pull system to conduct the experiment because it allowed the embryos in the liquid to flow through the channel. The microfluidic flow control device served to perform the push-pull system. The positive system provides a pushing force from an air compressor with a positive pressure (0–345 mbar) to drive the oil and medium into the chip. In contrast, the negative system generates a pulling force from an air pump with a negative pressure (0 to −345 mbar) to extract the waste liquid from the chip (see Figure 3).

We defined the time when the embryo fertilized as Day 0. The two-cell stage embryo was used to inject into the channel as a start of the experiment on Day 1.5 because it is difficult to detect whether the zygote is fertilized or not. The medium (20 μL) was put with three two-cell embryos on top of the embryo inlet on Day 1.5. It was then pulled into the generating area and cut into small droplets of size about 0.7 μL. Some droplets encapsulated an embryo, but most did not. We retained a droplet with an embryo and abandoned a droplet without embryo in the sorting area. The volume of the selected droplets was enlarged to 5 μL in the enlargement area. From Day 1.5 to 3.5, the embryos were cultured in the droplets with a slow flow in the storage area. On Day 3.5, the medium of size 4 μL was extracted from each droplet through the replacement area and the extracted medium was recycled for another examination. The droplet was enlarged again to size 5 μL with the new medium. From Day 3.5 to 5.5, the embryos were cultured in the droplets again in the storage area. On Day 5.5, the 4-μL medium was extracted from the droplet. These steps are shown in Figure 4.

In the experiment, we set the flow rate of the oil inlet (i.e., *Q*_oil inlet_) from 100 µL/h to 500 µL/h and the total flow rate of two outlets (i.e., *Q*_two outlets_) from *Q*_oil inlet_ + 100 µL/h to *Q*_oil inlet_ + 500 µL/h. As indicated in Figure 5a, the volume of the droplet increased with *Q*_two outlets_ when *Q*_oil inlet_ was fixed. In Figure 5b, the interval between two droplets decreased when the flow rate of two outlets increased at any fixed *Q*_oil inlet_. In Figure 5, the frequency of droplet generation decreased when the flow rate of the two outlets increased at any fixed *Q*_oil_ inlet.

## 3. Results and Discussion

Through our experimental experience, we found that a droplet of volume less than 1 µL, with an interval greater than 0.5 mm between each two droplets, and a frequency of generation about two droplets per minute were the best parameters. Exceeding the range of flow rate can result in a merger between droplets. While enlarging the droplet size to 5 µL, if two droplets are close enough, there is the possibility that the droplets will merge to prevent this from happening; time interval and flow rate between droplets are important parameters to be considered for this experiment. We chose a positive flow rate of 400 µL/h at the oil inlet and a total outlet flow rate of 600 µL/h to generate the desired droplet. The actual experimental results are shown in Figure 6a, in which a bead was encapsulated in a droplet.

### 3.1. Setup of Droplet Generating

To test the droplet generation, we set the flow rate of oil inlet *Q*_oil inlet_ from 100 µL/h to 500 µL/h and the total flow rate of two outlets *Q*_two outlets_ from *Q*_oil inlet_ + 100 µL/h to *Q*_oil inlet_ + 500 µL/h. As indicated in Figure 6a, the volume of the droplet increased with *Q*_two outlets_ at the fixed *Q*_oil inlet_. The frequency of droplet generation also increased with *Q*_two outlets_, as indicated in Figure 6b. The interval of each droplet decreased with increased *Q*_two outlets_. We found that the optimal values of droplet volume, interval between droplets, and frequency were <1 µL, >0.5 mm, and 2 droplets/min, respectively. On the basis of this result, we chose a positive flow rate of 400 µL/h at the oil inlet and a total outlet flow rate of 600 µL/h to generate suitable droplets (see Figure 6). 

### 3.2. Setup of Droplet Sorting

Droplet generation and sorting was conducted simultaneously; *Q*_oil inlet_ hence had value of 400 µL/h. When a droplet without an embryo flew through the sorting area, the flow rates of the waste area outlet and replacement area outlet were 550 μL/h and 50 μL/h, respectively. This droplet would flow to the waste area outlet. The flow rate of the replacement area outlet was non-zero so as to prevent the selected droplets from being pulled toward the waste area outlet. In contrast, when a droplet with at least one embryo flew through the sorting area, the flow rates of the waste area outlet and replacement area outlet were adjusted to 0 μL/h and 400 μL/h, respectively, which resulted in a droplet not being generated. After the droplet was pulled upwards, the parameters were adjusted to the previous condition to allow the next droplet to flow to the left side (all steps refer to Figure 6b).

### 3.3. Setup of Droplet Enlargement

After droplet sorting, *Q*_oil inlet_ had a value of 400 μL/h to push the selected droplets to the enlargement area. We stopped the pushing of the liquid from the oil inlet when the selected droplet reached the T-junction structure of the enlargement area. At the same time, the enlargement area inlet started to inject the HTF medium with flow rate of 300 μL/h into the selected droplet. After the droplet attained 5 μL, we terminated the injection from the enlargement area inlet, and turned on the oil inlet to push the enlarged droplet off the T-junction structure and allow the next droplet to flow in (Figure 7). To ensure that the droplet attained 5 μL, we let the difference between the target volume (2–10 μL) and original volume (0.7 μL) be divided by the flow rate of injection, 300 μL/h. The injection time of each target volume was then calculated. According to Figure 8, the error between target volume and experimental volume was less than 10%. We can, thus, confirm that this experiment can make a droplet attain accurate volume.

### 3.4. Droplet Replacement

The oil inlet was input at flow rate 400 μL/h to push the enlarged droplets into the replacement area after two days of culturing in the storage area. We then stopped pushing it when the enlarged droplet reached the T-junction structure of the replacement area. At the same time, the replacement area outlet was used to absorb the medium in the 5-μL droplets at flow rate of 300 μL/h. After attaining volume of 1–2 μL, we stopped the absorption from the replacement area outlet, and pushed the replaced droplet off by the T-junction structure from the oil inlet. The next enlarged droplet then flows to the T-junction structure (Figure 9).

## 4. Conclusions

In our research, we achieved a technique wherein a chip could perform four main functions—droplet generation, sorting, droplet enlargement, and restoration—simultaneously. A droplet was generated with volume in range of microlitres, whereas in past studies, the droplet was picolitre or nanolitre in size. Second, we undertook sorting of the droplets. A droplet carries an embryo, proceeds towards the enlargement area, and another droplet without embryo is abandoned and directed towards the waste area outlet using a manual sorting method. The next step involves enlarging the size of the selected droplet from 1 μL to 5 μL by injecting HTF or KSOM medium into the selected droplet, whereas during droplet generation, the size of the droplet is approximately 0.7–1 μL, although the droplet size is not enough for embryo growth; therefore, the droplet size is increased to 5 μL in the enlarging area. In the last step of the experiment, we removed the old HTF or KSOM medium from the enlarged droplet without losing embryo particles of diameter 100 μm. In subsequent steps, an embryo culture experiment is conducted. In future, we aim to conduct embryo culture from Day 1 to Day 5 and from Day 3 to Day 5.

## Figures and Tables

**Figure 1 micromachines-10-00756-f001:**
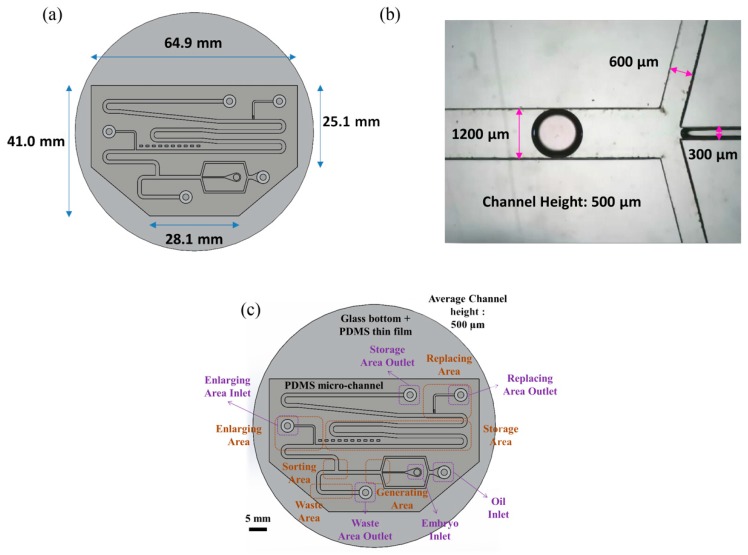
(**a**) The shape of the chip is hexagonal. The length of the longest side is 64.9 mm; the longest base length is 28.1 mm. (**b**) The width of our main channel is 1200 μm; the height of the channel is 500 μm. (**c**) There are six sections and six entrances in the chip.

**Figure 2 micromachines-10-00756-f002:**
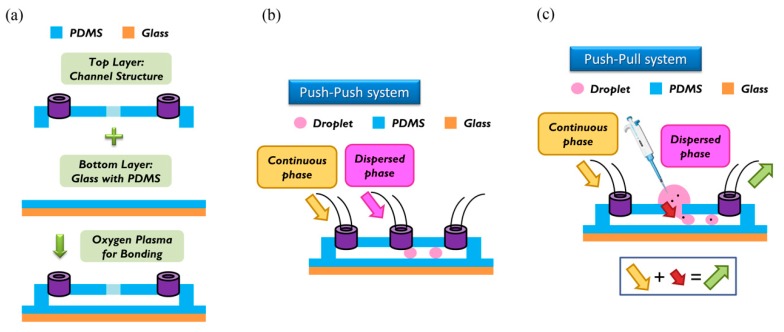
(**a**) Top layer (channel structure) and bottom layer (glass with polydimethylsiloxane (PDMS)) were bonded together with an oxygen plasma. (**b**) In the push-push system, continuous phase and dispersed phase are both pushed into the channel. (**c**) In the push-pull system, a continuous phase is also pushed into the channel, but the dispersed phase is pulled by the outlet behind. In our research, we used the push-pull system to load particles into the channel.

**Figure 3 micromachines-10-00756-f003:**
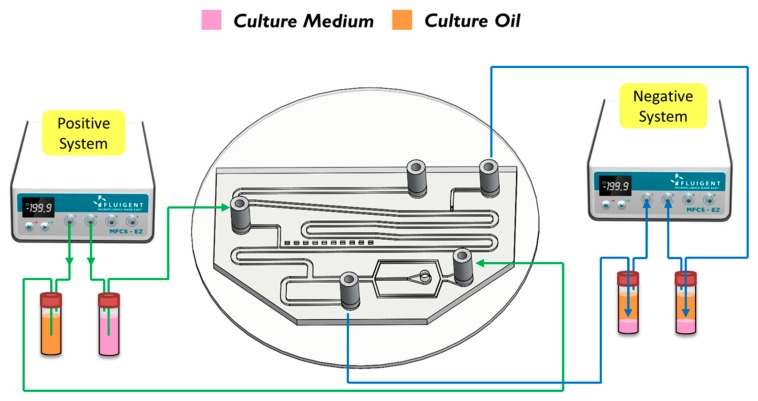
A positve system pushed the oil and the medium into the chip. In contrast, a negative system pulled the waste liquid from the chip.

**Figure 4 micromachines-10-00756-f004:**
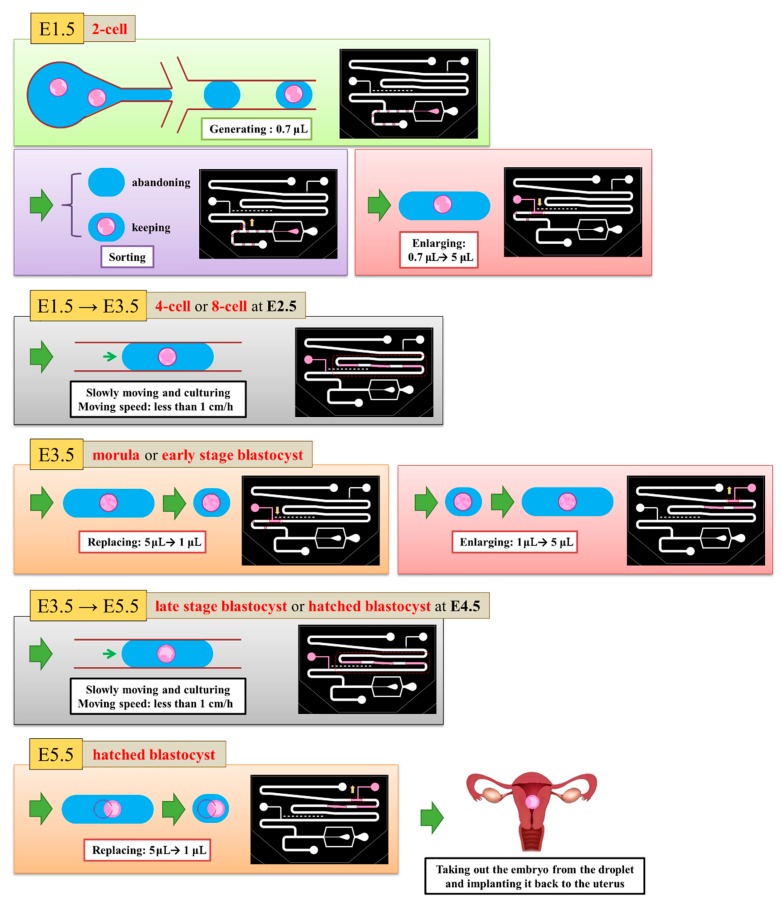
5.5 days are required to culture an embryo from zygote to hatching. At Day 1.5, we generated 0.7-μL droplets and retained the droplets with embryos. We enlarged all selected droplets to 5 μL. From Day 1.5 to Day 3.5, the embryos grew in the 5-μL droplets. At Day 3.5, we absorbed the liquid from the droplets until the size became 1 μL. We then enlarged all droplets to 5 μL. From Day 3.5 to Day 5.5, embryos grew in the 5-μL droplets again. At Day 3.5, we absorbed the liquid from the droplets until the size attained 1 μL to finish our experiment.

**Figure 5 micromachines-10-00756-f005:**
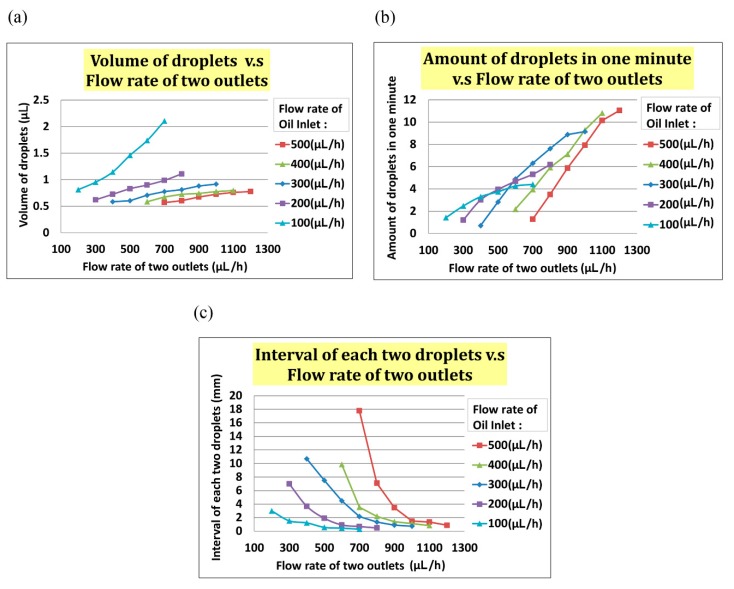
(**a**)The volume of a droplet increased when the total flow rate of the two outlets increased. (**b**)The frequency of droplet generating increased when the total flow rate of the two outlets increased. (**c**)The interval between two droplets increased when the total flow rate of the two outlets increased.

**Figure 6 micromachines-10-00756-f006:**
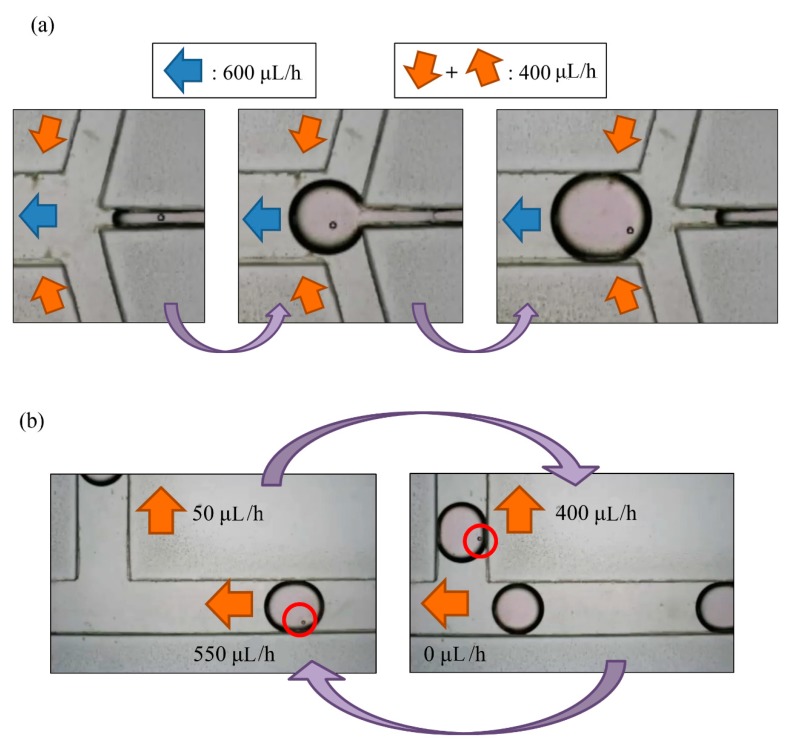
(**a**) A droplet with a PS bead was generated in the generating area; (**b**) the total flow rate of outlets was 600 μL/h for droplet generation. When a droplet with a particle reached the sorting area, we adjusted the flow rate of the waste area outlet from 550 μL/h to 0, and the flow rate of the replacement area outlet from 50 μL/h to 400 μL/h. When the droplet moved upwards, we readjusted the parameters.

**Figure 7 micromachines-10-00756-f007:**
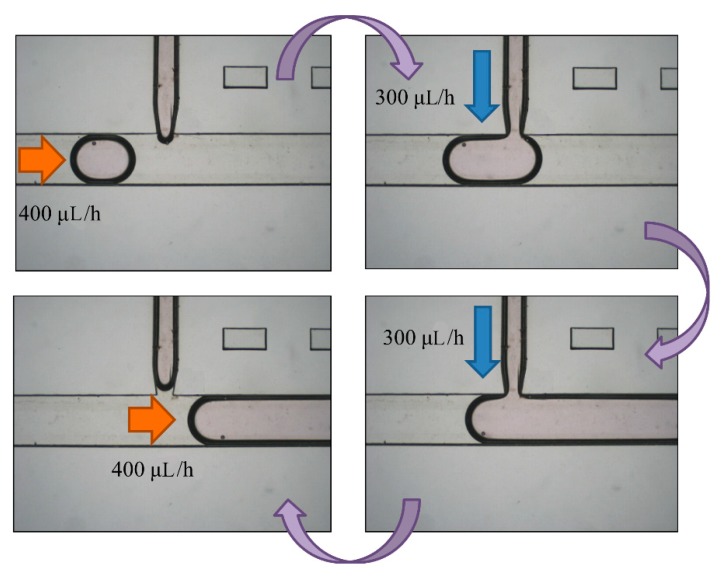
A selected droplet was pushed to the enlargement area with an oil inlet at flow rate of 400 μL/h. We turned off the oil inlet and opened the enlargement area inlet to inject more medium to the droplet at 300 μL/h. We eventually stopped the injection and opened the oil inlet to cut off the enlarged droplet.

**Figure 8 micromachines-10-00756-f008:**
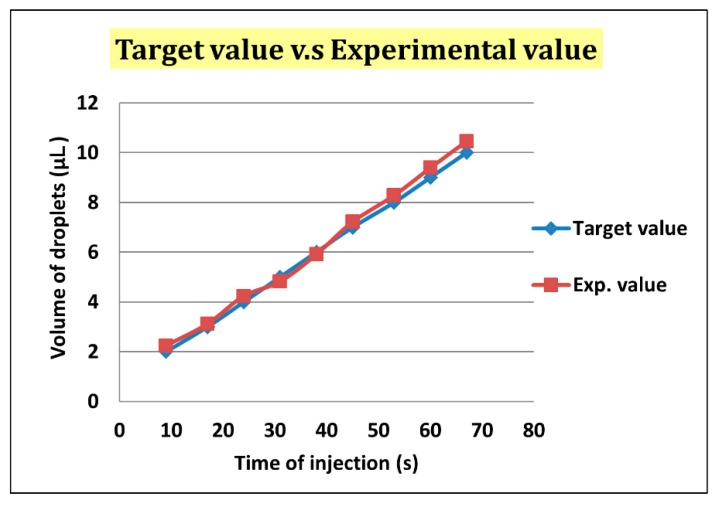
The error of the target volume and the experimental volume was less than 10%.

**Figure 9 micromachines-10-00756-f009:**
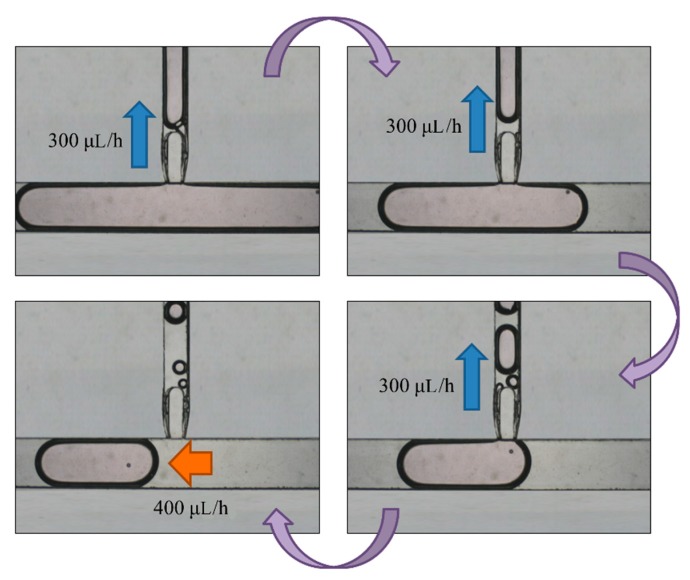
An enlarged droplet was pushed to the replacement area by the oil inlet at a flow rate of 400 μL/h. We then turned off the oil inlet and opened the replacement area inlet to absorb the medium from the droplet at 300 μL/h. Finally, we stopped the absorption and opened the oil inlet to cut off the droplet. The 100-μm bead was not taken out because of the blocking of the fork-shaped channel.

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
