# Peer review of "Unveiling the Potential of Droplet Generation, Sorting, Expansion, and Restoration in Microfluidic Biochips"

_micromachines, 2019, doi:10.3390/mi10110756_

Round 1
Reviewer 1 Report
comments attached

Reviewer 2 Report
A hybrid microfluidic device for cell culture was developed in this study. The chip has a droplet generation nozzle and a few other inlets to add cell medium to the droplet. This topology may have strength for cell sorting and long-term cell culture, however, there are some major technical and ethical issues of the application suggested by this article.
Technical issues:
The title does not depict the theme of the article. From the readers’ point of view, the title is for a review article. Page 2, the first paragraph says the traditional fabrication process is tedious but there is no innovation in the fabrication process presented in this study. The PDMS chip is bonded to a thin layer of PDMS on a glass piece. The bonding strength of PDMS-glass is much stronger than PDMS-PDMS, why a layer of PDMS is sandwiched in? In the ‘Conclusion’ section, the statement about the droplet size in microfluidic channels is not accurate. Using large droplets in microfluidic channels is not an unprecedented attempt at all. This should not be given credit in the conclusion section. Evaporation of the medium is significant for 3-5 days of cell culture. However, this problem is not mentioned in the article. In the conclusion section, the last sentence, why pick up these experimental periods? There is no explanation and references for this statement. The rationale for using this device for human samples is absurd. As presented in this article, it looks unnecessary and insignificant.Ethical:
The article lacks ethical statement regarding embryo studies. Sterilization for the entire experiment is absent in this article. It should be banned to use this non-sterile device to conduct experiments with human samples.English issues:
Proofread by native English speakers is strongly recommended for the revised version in the future.
Reviewer 3 Report
The paper shows good practicality ( I assume – as there is lots of missing information) but is not at all well organized and the findings are not clearly presented.
Since 1990, when A. Mantz, N. Graber and H.M. Widmer published in Sensors and Actuators the paper with the title: “Miniaturized total chemical analysis systems: A novel concept for chemical sensing”, Microfluidics start to pick concepts and ideas that brought the field at the present state. Please check the paper and see the way that paper is written; it entertains and attract researcher in the field.
The paper presents a microfluidic device which is poorly described and practically, impossible to reproduce. The presented pictures are of very low quality – in my view that cannot be published in the present form.
The scope of the work is, I assume, of great importance but this has not been started anywhere in the text. Why this complex process needs to be completed? Why the eggs cannot grow while outside of the oil droplet?
What is the actuation – I understood that is pneumatic but I might be wrong. These specs are not revealed and to make the paper suitable for publication, such information needs to be provided in the text.
I tried to understand the configuration of the microfluidic circuit. I could not make sense as there is no drawing (the pictures are very poor quality) of the circuit with the explanation of what part is doing what. I assume the circuit evolved from the first design to the last. The evolution was based on observations that re not presented.
The paper needs to be completely re-written to be considered for publication in Micromachines.
The new paper has to be written bearing in mind that the scope of a publication is to provide the information to other groups within the community who could continue the work of pushing forward the field of knowledge.
Round 2
Reviewer 2 Report
I am fine with the current form if 'Embryo Culture' can be included in the title.
Reviewer 3 Report
Congratulation for the published paper in Micromachines!